# Markovnikov hydroamination of terminal alkenes by phosphine redox catalysis

Flora Fan[1], Kassandra F. Sedillo[2], Alexander J. Maertens[1] & Abigail G. Doyle[1✉]

Main-group catalysts that mimic transition metal reactivity can expand substrate tolerance and enable transformations not possible at present with metal catalysis[1]. The discovery that P$^{III}$ and P$^V$ phosphorus intermediates can undergo transition-metal-like two-electron chemistry raises the question of whether radical P$^{IV}$ intermediates can mimic other elementary steps in organometallic chemistry[2,3]. Here we describe a phosphine–photoredox catalyst system that promotes intermolecular Markovnikov hydroamination of unactivated terminal alkenes with numerous classes of N–H azoles, a reaction that is not possible with late transition metal catalysis. Experimental and computational mechanistic studies support a new elementary step for main-group catalysis, in which a phosphine radical cation activates the alkene to nucleophilic amination by the azole, a step otherwise associated with transition metals. Given the broad value of nucleophilic alkene functionalization in transition metal catalysis, this P$^{IV}$ mechanism could offer new opportunities for main-group element catalysis and chemical synthesis.

Catalytic methods to construct carbon–nitrogen (C–N) bonds are widely sought because of the prevalence of amine functionality in molecules in the biomedical, agrochemical and fine chemical industries. Hydroamination, the formal addition of an N–H bond across an unsaturated C–C bond, is a particularly valuable transformation for C($sp^3$)–N bond formation because it exploits the abundance and structural variety of alkene and amine starting materials to form products with high atom economy under redox-neutral conditions[4,5]. However, these reactions do not occur without a catalyst. Late transition metal (TM)-catalysed alkene hydroamination has been extensively developed[6–8], taking advantage of different elementary steps involving a metal and alkene, such as (i) migratory insertion; (ii) nucleophilic attack on a metal–alkene complex; or (iii) metal-catalysed hydrogen atom transfer (MHAT)[9,10] (Fig. 1a). Despite the numerous catalysts identified and synthetic advances resulting from their development, a general solution to catalytic intermolecular, Markovnikov hydroamination of unactivated, terminal alkenes is not available[8], and only limited examples exist for this reaction class using azoles as nitrogen sources[11–13]. As unactivated terminal alkenes are produced on manufacturing scale and comprise a substantial portion of the feedstock[14,15], and azoles are widely represented in medicinal chemistry, materials science and agrochemistry[16], they both offer underutilized and attractive substrate classes for C($sp^3$)–N bond formation through hydroamination.

In 2014, the Hartwig group reported that an Ir catalyst promotes the addition of indoles to terminal aliphatic alkenes by turnover-limiting migratory insertion to give hydroamination products with Markovnikov selectivity[11]. More recently, the Akai and Zhang groups described an MHAT approach to Markovnikov-selective hydroaminations between terminal aliphatic alkenes and benzotriazoles using Co catalysis, in which only singular examples of other azoles, tetrazole and benzimidazole were demonstrated[12,13]. The specificity of these solutions to a singular azole class can be attributed to the propensity of TMs

to undergo deactivation by azole coordination[17,18] or unproductive side reactions, such as azole oxidative addition[19,20]. More generally, progress in this area has been impeded by the fact that unactivated, terminal alkenes are weak ligands for TMs, and side products arising from alkene isomerization commonly outcompete the desired, often thermally neutral, transformation[6–8,21].

Recently, researchers have sought to mimic elementary steps of TMs using abundant main-group elements and develop main-group catalysts that expand reactivity in synthetic reactions that are challenging for TM catalysis[1,2,22,23]. Organophosphorus derivatives have been shown to undergo two-electron reduction and/or oxidation, oxidative addition, ligand exchange and reductive elimination in stoichiometric contexts[24–26], and have emerged as versatile redox-active main-group catalysts for C–F and C–N bond formation (Fig. 1b)[27–30]. Apart from representing sustainable alternatives to late TMs, organophosphorous derivatives often show complementary functional group compatibility, such as to Lewis-basic substrates. With this in mind, we questioned whether it might be possible to mimic the elementary steps involved in TM-catalysed nucleophilic alkene functionalization with a phosphine catalyst, and in doing so, develop a generally applicable catalytic method for intermolecular, Markovnikov-selective hydroamination of azoles and unactivated terminal alkenes (Fig. 1c).

Here, we report a cooperative phosphine–photoredox catalyst system that achieves Markovnikov hydroamination between a broad range of N–H azoles and terminal, aliphatic alkenes (Fig. 1d). Our experimental and computational mechanistic studies indicate that phosphine radical cations promote nucleophilic amino-phosphination of alkenes by two competing, energetically feasible pathways, (ia and ib) and (ii) in Fig. 1c, which mimic inner- and outer-sphere mechanisms for TM-catalysed alkene functionalization in industrially important reactions, such as the Pd-catalysed Wacker process and Pd-catalysed alkene amination[31,32]. However, because the phosphine-catalysed mechanism is initiated by

[1]Department of Chemistry and Biochemistry, University of California, Los Angeles, CA, USA. [2]Department of Chemistry, Princeton University, Princeton, NJ, USA. ✉e-mail: abigaildoyle@g.ucla.edu

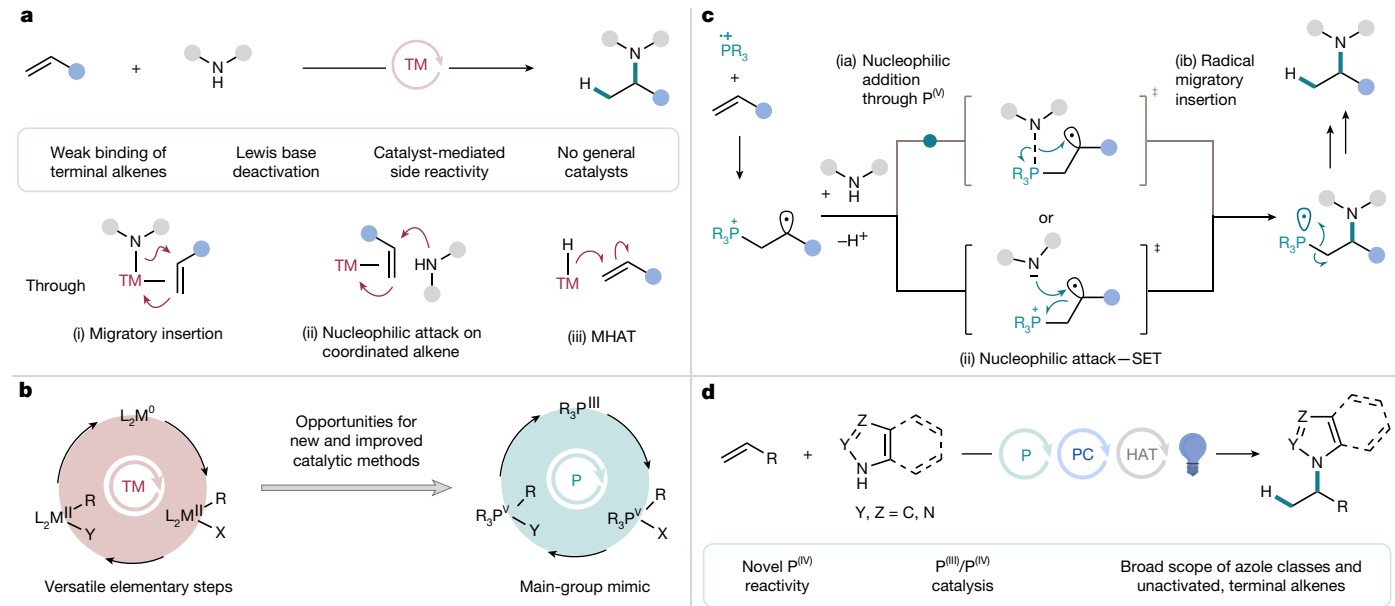

**Fig. 1 | Introduction. a**, TM-catalysed nucleophilic amination with terminal alkenes and limitations (red box). MHAT, metal-catalysed hydrogen atom transfer. **b**, Mimicking TM catalysis with phosphorus. **c**, Unexplored activation modes in phosphorus catalysis. SET, single electron transfer. **d**, Phosphine-catalysed Markovnikov-selective hydroamination with terminal alkenes.

an open-shell P[IV] species, polar addition of the nucleophile is accompanied by either a radical migratory insertion or an intramolecular electron transfer into the adjacent P–C anti-bonding orbital, in which the latter represents an unusual example of the microscopic reverse of a spin-centre shift process[33]. Subsequent functionalization of the resulting M–C bond proceeds by a one-electron (M = P) rather than two-electron (M = TM) mechanism, which is also distinct from most TM-catalysed nucleophilic alkene functionalization reactions. We anticipate that these similarities and differences will offer new opportunities for the design of synthetic transformations using phosphine catalysis.

## Reaction development

Previously, our laboratory reported that PCy₃ catalyses the hydroamination of alkenes with primary sulfonamides[34] and in collaboration with the Knowles group, hydroamination of alkenes with N–H azoles[35], both mediated by a visible light photoredox co-catalyst in a P[(III/IV)] cycle. These reactions proceeded with anti-Markovnikov selectivity, consistent with a mechanism involving the generation of a nitrogen-centred radical that undergoes anti-Markovnikov addition to the alkene. During our recent exploration of a low-performing substrate combination, the reaction of 3-phenylpyrazole with methylene cyclopentane, we discovered that use of P(p-OMePh)₃ as a catalyst unexpectedly afforded a complete switch in regioselectivity to Markovnikov hydroamination. As this regioselectivity outcome is prevalent for late TM-catalysed alkene hydrofunctionalization reactions, we were intrigued by the underlying reaction mechanism and its synthetic implications. To this end, we sought to optimize the reaction for unactivated, terminal alkenes, explore its synthetic scope and understand the phosphine-dependent change of mechanism for C(sp³)–N bond formation.

With 1-hexene as the alkene partner, we found that using 20 mol% of either P(p-OMePh)₃ or PPh₃, 2 mol% of [Ir(dF(Me)ppy)₂(dtbbpy)]PF₆ photocatalyst and 10 mol% of 2,4,6-triisopropylbenzenethiol (TRIP-SH) catalyst with 450 nm LEDs gave high yield and exclusive Markovnikov selectivity for the intermolecular hydroamination with 3-phenylpyrazole (1) to form N-alkylated product 2 (Extended Data Table 1, entries 1 and 2). Reducing the phosphine catalyst loading to 10 mol% significantly hindered activity (Extended Data Table 1, entry 3).

Notably, other triaryl and alkylphosphine catalysts were unreactive (Supplementary Table 3). Increasing the thiol loading led to reduced yield of product 2 (Extended Data Table 1, entries 4–6) with concomitant increase in formation of phosphonium by-product **B1**. Given the unique regioselectivity outcome of the hydroamination, we questioned whether trace TM may be present. Including 10 mol% of Pd(OAc)₂ or other metal salts such as Cu(OAc)₂, NiBr₂ or Fe(OTf)₃ (Extended Data Table 1, entry 7) as an additive completely shut down reactivity. Furthermore, control studies indicated no reactivity without light (entry 11), which is inconsistent with the presence of trace amounts of a TM impurity serving to catalyse the reaction. Although using only 1 equivalent (equiv.) of alkene led to reduced yield at 18 h (entry 12), the yield could be restored by allowing the reaction to run for 48 h (entry 13), demonstrating the utility of the method for Markovnikov hydroamination of more valuable alkene coupling partners. Notably, most late TM-catalysed hydroamination reactions with aliphatic terminal alkenes require a large excess of reaction partner (>5 equiv.) due to the weak binding of these substrates, regardless of the amine identity[8,36,37].

## Azole and alkene scope

Next, we sought to evaluate the scope of N–H azoles tolerated in the reaction (Fig. 2). We began our exploration with substituted pyrazoles, as this heterocycle class is one of the 10 most common in drugs approved by the US Food and Drug Administration[16]. Apart from 3-phenylpyrazole, 3-carboethoxypyrazole (3) is a competent substrate, undergoing Markovnikov-selective hydroamination in 78% yield. Although a pyrazole bearing a boronic ester substituent reacted in 36% yield (4), the utility of boronate esters for further derivatization by cross-coupling indicates the moderate success of this substrate is nonetheless a useful advance. Fomepizole (5), an alcohol dehydrogenase inhibitor, was N-alkylated in 57% yield. 4-Phenylimidazole (6), belonging to another highly prevalent heterocycle class in medicinal chemistry, underwent hydroamination in good yield. Biologically active compounds with the imidazole core, such as the neurotransmitter histamine (7) and the sedative dexmedetomidine (8) reacted with exclusive N-site selectivity.

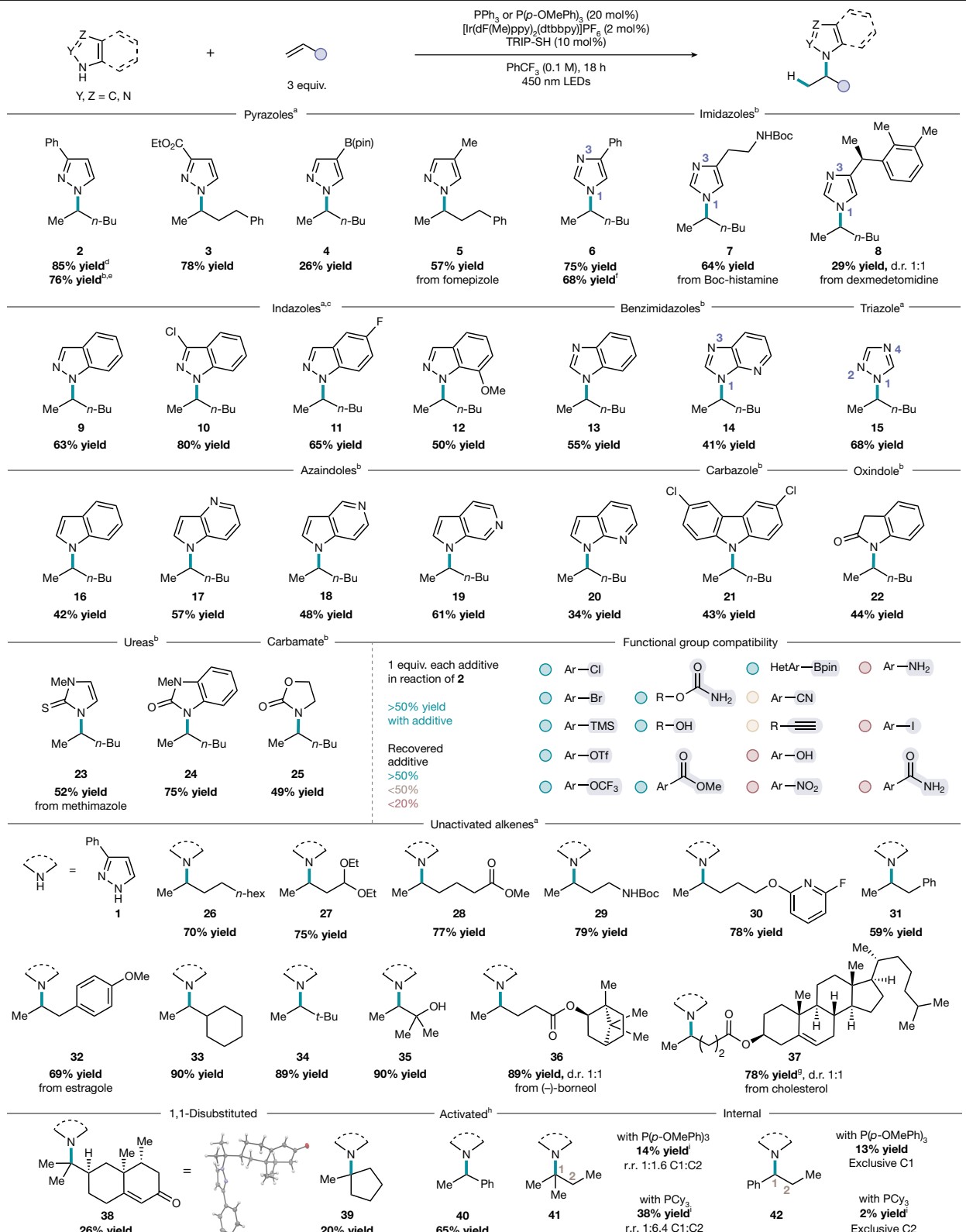

**Fig. 2 | Azole and alkene scope.** Reactions were performed on a 0.5 mmol scale using 1.0 equiv. of azole and 3.0 equiv. of alkene, PPh₃ or P(*p*-OMePh)₃ (20 mol%), [Ir(dF(Me)ppy)₂dtbbpy]PF₆ (2 mol%), and TRIP-SH (10 mol%) irradiating with 450 nm LEDs for 18 h. Isolated yield reported as an average of two runs. [a]PPh₃ was used as the phosphine catalyst. [b]P(*p*-OMePh)₃ was used as the phosphine catalyst. [c]Reaction run at 0.05 M with 2 × 427 nm Kessil lamps. [d]Isolated yield from reaction performed on 5 mmol scale. [e]Reaction run with P(*p*-OMePh)₃ (20 mol%), analytical ¹H NMR yield determined by comparison to

an internal standard of 1,3,5-trimethoxybenzene, reaction set up on the bench, sparging with N₂ for 5 min followed by addition of alkene, instead of in a glovebox. [f]Analytical ¹H NMR yield determined by comparison to an internal standard of 1,3,5-trimethoxybenzene, reaction set up under Schlenk technique instead of in a glovebox. [g]Reaction run with 2.0 equiv. of alkene. [h]Reaction run with P(*p*-OMePh)₃ (20 mol%), 1.2 equiv. of alkene and TRIP-SH (20 mol%). [i]Analytical ¹H NMR yield determined by comparison to an internal standard of 1,3,5-trimethoxybenzene.

Fused bicyclic heterocycles that contain multiple sites (at nitrogen(s) or carbon) on both rings for possible alkylation were also amenable to hydroamination under the same set of conditions. For example, unsubstituted indazole (**9**); 3-chloroindazole (**10**), which contains a halide handle for downstream cross-coupling; and 5-fluoroindazole (**11**), which provides a medicinally relevant fluorine substituent[38], underwent hydroamination in good yield. Despite its steric hindrance, 7-methoxyindazole afforded 50% yield of product **12** under the standard reaction conditions. In the PCy$_3$-catalysed anti-Markovnikov-selective hydroamination[35], 4-azabenzimidazole favoured the *N*3-alkylated isomer using methylene cyclopentane as an alkene partner (*N*1:*N*3 regioisomeric ratio 1:1.2). By comparison, 4-azabenzimidazole results in exclusively the *N*1-alkylated isomer **14** under these Markovnikov-selective conditions using P(*p*-OMePh)$_3$ as a catalyst, pointing to a distinct mechanism for C–N bond formation.

Incorporating another nitrogen atom in the form of 1,2,4-triazole (**15**) did not affect *N*-site selectivity, and the reaction proceeded with exclusive *N*1-alkylation. The indole core (**16**) is abundant in bioactive compounds, and 4-, 5-, 6- and 7-azaindoles (**17**–**20**) were all competent substrates under the catalytic conditions, providing a useful handle to perform nitrogen scanning in medicinal chemistry library campaigns[39]. Carbazole (**21**), oxindole (**22**), a partially saturated N–H azole, thiourea (**23**) and benzourea (**24**), and fully saturated carbamate (**25**) were also reactive in this hydroamination method, demonstrating that nitrogen nucleophiles other than unsaturated azoles are compatible. Although the standard reaction conditions call for the use of a glovebox, setting up the reaction under ambient conditions followed by N$_2$ sparging or using the standard Schlenk technique demonstrated similar reactivity; products **2** and **6** were obtained in 76% and 68% yield with N$_2$ sparging and the Schlenk technique compared with 85% and 75% yield using the glovebox, respectively. Furthermore, the model reaction with 3-phenylpyrazole proved scalable in batch to gram-scale (see Supplementary Information section 5 for further details), demonstrating the potential utility of this photocatalytic protocol for industrial applications. An additive screen showcased aryl chlorides, bromides and triflates to be compatible under the reaction conditions, demonstrating complementary functional group tolerance to TM-catalysed methods. Additives containing functional handles susceptible to oxidation or reduction under photochemical conditions (for example, aniline, phenol) were unsurprisingly less tolerated in the reaction.

Next, we surveyed the scope of terminal alkenes (Fig. 2). Unactivated monosubstituted terminal aliphatic alkenes such as 1-decene (**26**) proceeded with good yield, and various polar functional groups, including acetals (**27**), esters (**28**) and Boc-protected amines (**29**) were well-tolerated, providing additional handles for downstream derivatization of the products. Lewis-basic heterocycles, such as a pyridine moiety (**30**), did not affect reactivity. Allylbenzene, a substrate that is susceptible to alkene isomerization with TM catalysts[40], led to a single *N*-alkylation product (**31**) under the phosphine-catalysed reaction conditions. Furthermore, the structurally similar estragole, a naturally abundant phenylpropene, afforded the hydroaminated product (**32**) in 69% yield.

Moreover, terminal alkenes bearing di-substitution at the allylic position, such as vinylcyclohexane (**33**), proceeded with high yield under these phosphine–photoredox conditions. 3,3-Dimethyl-1-butene (**34**), which may undergo a 1,2-methyl shift to a more stable carbocation once coordinated to a TM centre, underwent hydroamination in excellent yield. Whereas alcohols could serve as competing nucleophiles, a tertiary alcohol was also compatible, probably because of its steric encumbrance, and afforded a privileged 1,2-aminoalcohol scaffold (**35**). Bioactive alkenes derived from natural products and steroids, such as (L)-borneol (**36**) and cholesterol (**37**), gave a 1:1 mixture of diastereomeric *N*-alkylated products in high yield, and hydroamination was selective for the terminal rather than internal alkene. Despite the

inclusion of a nucleophilic phosphine catalyst, the insect repellent nootkatone (**38**) could also be used chemoselectively for reaction with the terminal alkene, bypassing potential competitive Baylis–Hillman reactivity with the enone and delivering a highly congested C–N bond. The low yield likely results from 1,1-di-substitution of the starting alkene, as also illustrated by (**39**), suggesting the steric profile of the alkene as an important factor in reactivity. Styrene (**40**), an activated terminal alkene, underwent Markovnikov hydroamination with 3-phenylpyrazole in 65% yield. Whereas internal alkenes (**41**) and (**42**) exhibited low reactivity, the hydroamination regioselectivity was strongly phosphine-dependent, with P(*p*-OMePh)$_3$ delivering greater Markovnikov or exclusively Markovnikov selectivity, respectively, compared with that obtained with PCy$_3$ in these case studies. Overall, this hydroamination method effectively couples together a range of N–H azoles and functionalized terminal alkenes with exclusive *N*-site selectivity and Markovnikov regioselectivity, encompassing substrates both compatible and incompatible with TM catalysis under a general set of conditions.

## Mechanistic investigation

Given the well-precedented use of TMs for Markovnikov-selective nucleophilic alkene functionalization, we proposed that PAr$_3$-catalysed Markovnikov selectivity could result from a mechanism analogous to metal catalysis, such as (i) migratory insertion; (ii) nucleophilic attack on a metal–alkene complex; or (iii) MHAT.

MHAT represents an alkene functionalization mechanism common to TM-catalysed Markovnikov transformations[12,13,41], such as the Mukaiyama hydration, that could be responsible for the phosphine-catalysed Markovnikov hydroamination reaction (Fig. 3a). As potential support for this proposal, the Studer group has reported a phosphine-promoted alkene hydrogenation through MHAT from a phosphoranyl radical **INT-1** (ref. 41). Adventitious water, or the azole, could serve as the source of a hydrogen atom in the hydroamination reaction, followed by C–N bond formation through oxidative radical polar crossover of the resulting secondary carbon radical **INT-2**. However, no hydrogenation by-products are observed (for example, hexane), even with increased thiol loading. Furthermore, the minor by-products detected (**2A** and **2B**) imply the intermediacy of a primary radical, which is incompatible with this mechanism. We also ruled out the MHAT mechanism on the basis that it would require reduction of the P$^V$ by-product[42]—a stoichiometric reagent in the Studer hydrogenation—and oxidation of the unactivated secondary radical **INT-2**, neither of which is feasible given the photocatalyst potentials[43].

We thus turned to nucleophilic alkene functionalization mechanisms (i) and (ii), which would both involve the intermediacy of a phosphine radical cation (**INT-3A**) (Fig. 3b, top). To evaluate these proposals, we sought evidence that the phosphine was the primary quencher of the excited-state photocatalyst. Stern–Volmer studies indicate that both PPh$_3$ and P(*p*-OMePh)$_3$ quench the excited-state photocatalyst (Supplementary Fig. 2). Therefore, phosphine radical cation (**INT-3A**) is likely a productive intermediate in Markovnikov-selective hydroamination. P(*p*-OMePh)$_3$, a more electron-rich phosphine, is required when using azole substrates that can undergo competitive oxidation by the excited-state photocatalyst (Supplementary Fig. 4). Alkenes are known to add to phosphine radical cations to form distonic radical cations (**INT-4**) with the positive charge primarily localized on phosphorus and the spin primarily localized on carbon[34,44]. Accordingly, radical functionalization of this intermediate at carbon is known in a limited but growing number of cases[29,45,46], most commonly by HAT or intramolecular radical cyclization onto the aryl substituents on phosphorous[44,47,48] (Fig. 3b, top). Our observation that increased thiol loading reduced product yield and increased phosphonium salt (**B1**) formation in the optimization studies (Extended Data Table 1, entries 4–6) is consistent with the intermediacy of a distonic radical

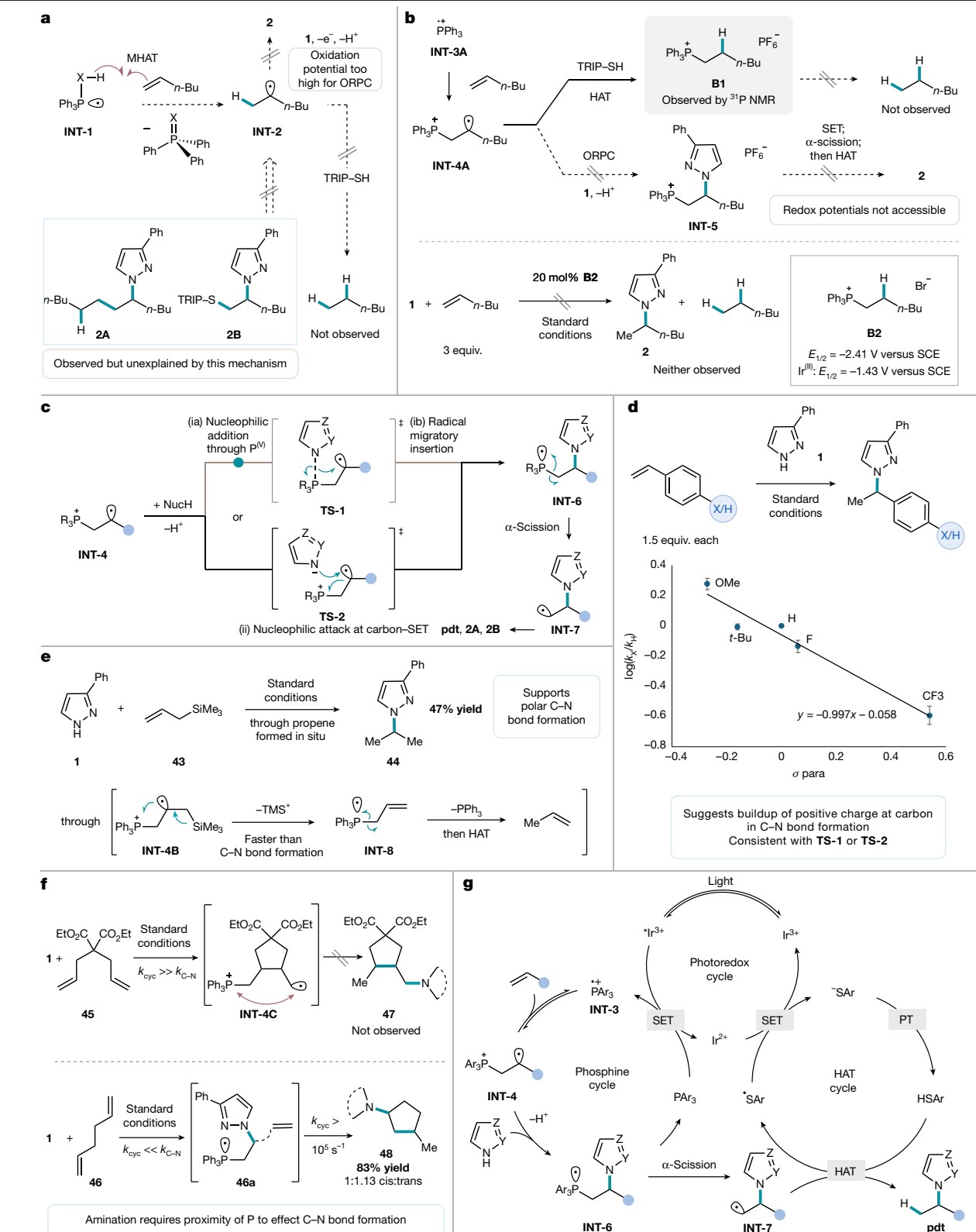

**Fig. 3 | Mechanistic studies. a**, Considering a MHAT mechanism; side products isolated and characterized. OPRC, oxidative radical polar crossover. **b**, Ruling out alternative reactivity pathways from distonic radical cation; reaction performed on a 0.1 mmol scale with [Ir(dF(Me)ppy)₂dtbbpy]PF₆ (2 mol%), TRIP–SH (10 mol%) and PhCF₃ (0.1 M), irradiating with 450 nm LEDs for 18 h, absence of products observed by ¹H NMR of reaction mixture. SCE, saturated calomel electrode. **c**, Proposed C–N bond formation involving nucleophilic amination through (ia) nucleophilic addition and (ib) migratory insertion or (ii) nucleophilic addition, intramolecular SET. **d**, Hammett relationship from competition studies

performed on a 0.1 mmol scale with 1.0 equiv. 3-phenylpyrazole and 1.5 equiv. each of styrene and *para*-substituted styrene; relative yields obtained in triplicate as a proxy for log(*k*ₓ/*k*ₕ) were determined by ¹H NMR spectroscopic analysis. **e**, Isolated yield on a 0.5 mmol scale; propene observed by ¹H NMR for a 0.1 mmol scale reaction in toluene-*d*8 in a J-Young tube. **f**, Radical cyclization experiments. Top, absence of *N*-alkylated product observed by ¹H NMR of reaction mixture; bottom, isolated yield on a 0.5 mmol scale with PPh₃ (20 mol%) as phosphine catalyst. **g**, Proposed mechanism for Markovnikov selectivity.

cation (**INT-4A**), which could undergo competitive HAT with TRIP-SH. Notably, monitoring a standard reaction by [31]P NMR showed that **B1** is not consumed under the reaction conditions, and hexane, a formally hydrogenated by-product that could result from C–P cleavage of the phosphonium, is not observed. Subjecting **B2**, a phosphonium salt analogue of **B1**, to the catalytic conditions in the absence of added phosphine led to no desired hydroamination product (Fig. 3b, bottom), indicating that neither direct P–C scission of **B2**, nor reduction of **B2** to its corresponding phosphoranyl radical followed by α-scission, are occurring under the reaction conditions. Reduction of the phosphonium to the phosphoranyl radical ($E_{1/2} = -2.41$ V compared with SCE) by Ir[(II)] ($E_{1/2} = -1.43$ V compared with SCE) or Ir[(III)*] ($E_{1/2} = -0.92$ V compared with SCE) is thermodynamically unlikely[43]. As a result, we concluded that phosphonium salts **B1**, and **INT-5**, which could arise from oxidative radical polar crossover and nucleophilic trapping from **INT-4A**, are not productive intermediates in C–N bond formation.

Hence, we propose that C–N bond formation proceeds from **INT-4** to generate phosphoranyl radical **INT-6**—implicating a nucleophilic amination of the distonic phosphine radical cation **INT-4**—an elementary step that is not known, to the best of our knowledge. Analogous to the pathways (i) and (ii) in which TMs catalyse nucleophilic alkene functionalization, two pathways can be considered for nucleophilic amination of the distonic radical cation **INT-4**: (ia) nucleophilic attack of azole at phosphorus to afford a P[V] intermediate, followed by (ib) radical migratory insertion to the carbon-centred radical through **TS-1** or (ii) direct nucleophilic addition–intramolecular SET between the azole and distonic radical cation (**TS-2**) (Fig. 3c). Pathway (ii) can be considered as the microscopic reverse of a spin-centre shift (SPS) process[33], in which addition of a polar nucleophile, the azole, pushes the odd electron into an adjacent anti-bonding orbital. Although the acceptor orbitals in most SPS processes are π* in character, this proposed mechanism suggests that similar chemistry is also possible with P–C anti-bonding orbitals. Subsequent α-scission from phosphoranyl radical **INT-6** to give terminal radical **INT-7** is kinetically facile and thermodynamically favourable ($\Delta G^{\ddagger} = +6.3$ kcal mol⁻¹, $\Delta G = -14.7$ kcal mol⁻¹; see Supplementary Fig. 44). Notably, the generation of by-products **2A** and **2B** is consistent with the intermediacy of **INT-7** and indicates that this activation mode could enable the development of a suite of regioselective alkene difunctionalization reactions.

We sought to experimentally evaluate the feasibility of the nucleophilic amination step based on physical organic reactivity principles. In TM-catalysed nucleophilic amination of alkenes, the carbon that undergoes C–N bond formation develops a partial positive charge following coordination to the metal centre. Under the hypothesis that a similar process could be occurring in our system by either of the nucleophilic amination pathways (i) or (ii), we investigated a potential linear free-energy relationship (LFER) through competition studies using substituted styrene partners and N–H azole **1** (Fig. 3d). An LFER was observed with $\sigma_p$ values, but not radical $\sigma$ values (Supplementary Figs. 33 and 34) with a $\rho$ value of −1.0. Similar $\rho$ values have been observed for TM-catalysed nucleophilic functionalization reactions of styrenes in relation to alkene coordination[49–52]. In our system, the alkene addition step to the phosphine radical cation, which would also exhibit a negative LFER, has been suggested to be fast and reversible[44,46]. Therefore, the negative $\rho$ value is more likely to be a readout of the rate-determining C–N bond formation step, consistent with build-up of positive charge at carbon in the bond-forming transition state (see below). Consistent with this step being rate-determining, an inverse secondary kinetic isotope effect ($k_H/k_D = 0.95$) was observed by competition experiment using styrene-α-$d_1$ (Supplementary Information section 9).

We proposed that if a nucleophilic amination is responsible for C–N bond formation, an alkene bearing a β-silicon group could interrupt C–N bond formation by an intramolecular elimination and electron transfer according to the β-silyl effect, providing further evidence for the polar nature of the C–N bond formation (Fig. 3e). In this case,

β-elimination of trimethylsilyl cation would be expected to afford phosphoranyl radical **INT-8** from **INT-4B**, which on α-scission and HAT, would generate propene in situ. If silyl radicals were to be eliminated instead, a phosphonium by-product would be detected, which could not undergo subsequent P–C cleavage. Under standard reaction conditions, reaction of N–H azole **1** with allyltrimethylsilane (**43**) afforded N-iso-propylated product **44** in 47% yield with observation of propene, consistent with the proposed nucleophilic functionalization of the distonic radical cation.

As a final test of the proposed C–N bond formation, we sought to evaluate radical cyclization substrates **45** and **46** (Fig. 3f). We proposed that if C–N bond formation is assisted by P—either in the stepwise (i) or concerted mechanism (ii)—radical cyclization substrates that displace the C-centred radical from the phosphonium before C–N bond formation would not afford aminated products. Specifically, with diethyl diallylmalonate (**45**), the proposed distonic radical cation would be expected to undergo fast intramolecular 5-exo-trig cyclization[53,54] to **INT-4C** before intermolecular C–N bond formation. As the carbon radical in **INT-4C** is located further from the cationic phosphorus, we expected that C–N bond formation would not take place. Consistent with this hypothesis, we observe no N-alkylated products, but phosphine is fully consumed according to [31]P NMR, likely to be phosphonium salt after competitive HAT. Conversely, with a radical cyclization substrate that would undergo slow radical cyclization from the distonic radical cation, but fast cyclization from the resulting primary radical post-α-scission, we would expect formation of N-alkylated product. With 1,5-hexadiene (**46**), we observed 83% yield of **48**, which we attribute to formation by a favourable 5-exo-trig cyclization ($k \approx 2 \times 10^5$ s⁻¹) (ref. 53) from the terminal radical that results from α-scission of **46a**.

We then turned to DFT to provide additional support for the feasibility of the nucleophilic amination step and further distinguish between a stepwise (ia) nucleophilic addition followed by (ib) migratory insertion or a (ii) direct nucleophilic addition–intramolecular single electron transfer (SET). Using P(p-OMePh)₃ as a catalyst, methylene cyclopentane as an alkene and 3-phenylpyrazole as an azole, we found a stable pentavalent phosphorane **INT-9** that proceeds through **TS-1** to phosphoranyl radical **INT-6A** (Fig. 4a, solid pathway), supporting the first pathway for nucleophilic amination. Comparison of the P–N and P–C bond lengths in **INT-9** and **TS-1** suggests an asynchronous migratory insertion, which led us to identify a transition state **TS-2** that would arise from pathway (ii), which has a higher activation energy but is still kinetically feasible ($\Delta G^{\ddagger} = +16.5$ kcal mol⁻¹, $\Delta G = -13.8$ kcal mol⁻¹) (Fig. 4a, dotted pathway). An NBO charge and spin analysis on **INT-9** and **TS-1**, and **INT-4D** and **TS-2**, respectively (Fig. 4b), shows a similar loss of electron density at carbon and an increase in radical spin density at phosphorus, consistent with our LFER (Fig. 3d), suggesting that either nucleophilic amination pathway could be operative.

Given the structural variety of azoles that are compatible in the reaction, we wondered if the likelihood of either pathway could be distinguished by substrate identity, with direct nucleophilic addition–intramolecular SET between the azole and distonic radical cation preferred for bulkier azoles. To interrogate this possibility further, we calculated the potential energy surface of this nucleophilic amination pathway for azabenzimidazole **14**. For the Markovnikov-selective conditions, we did not find a stable pentacoordinate **INT-9**-type intermediate for this substrate combination representative of pathway (i). However, transition states corresponding to direct nucleophilic addition–intramolecular SET were identified, representative of pathway (ii). To validate the feasibility of this pathway, we questioned whether our calculations could rationalize the complete N1-site selectivity observed experimentally. Under the proposed C–N bond formation manifold, the computed activation barriers for N1 compared with N3 alkylation have a $\Delta\Delta G^{\ddagger}$ of 4.9 kcal mol⁻¹ favouring N1 alkylation (Fig. 4c). Distortion–interaction analysis of the two transition states suggests a greater interaction between the azole and distonic radical cation for N1 nucleophilic attack, as well as reduced distortion from

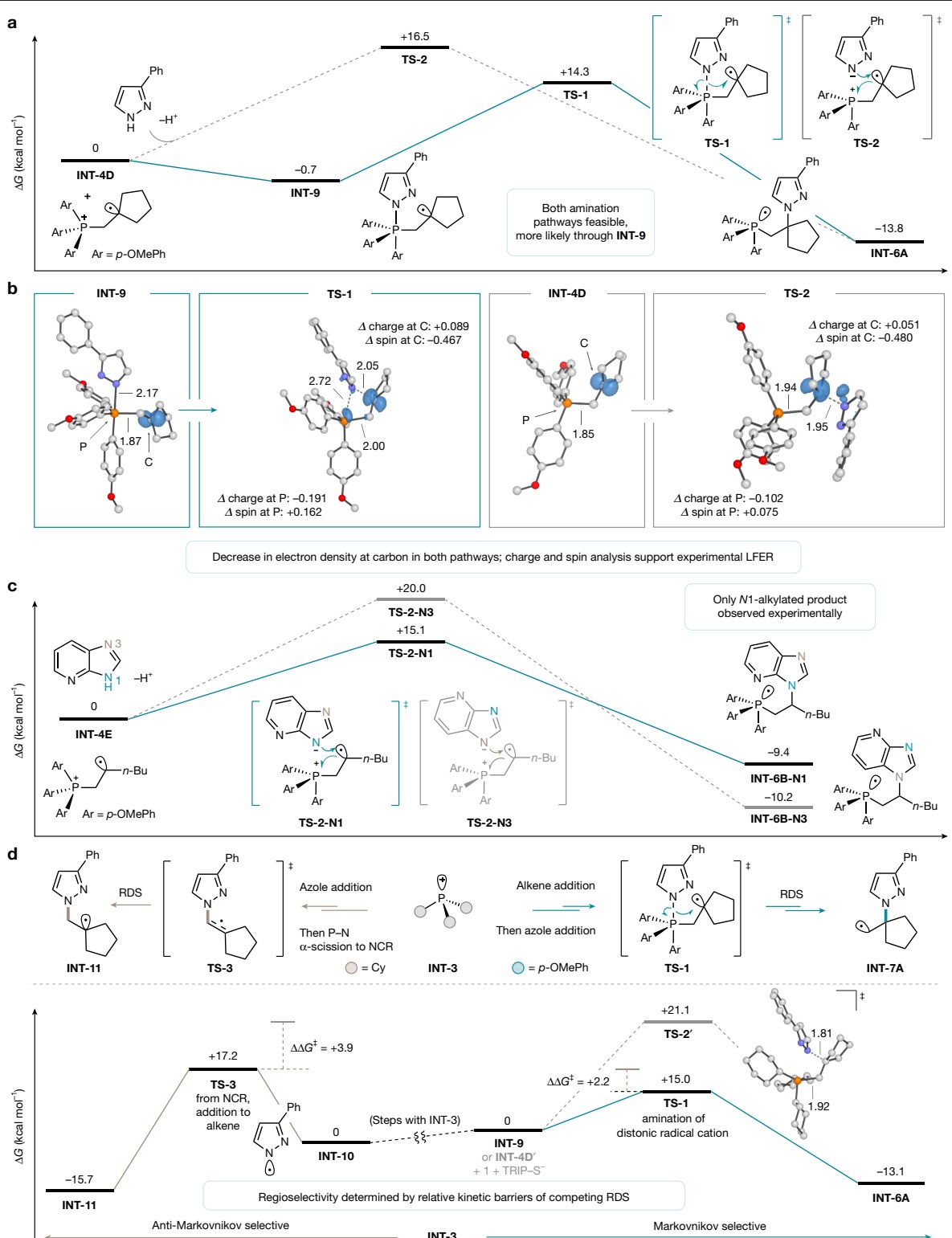

**Fig. 4 | DFT computational studies. a**, Reaction coordinate of proposed C–N bond formation with 3-phenylpyrazole, methylene cyclopentane and P(*p*-OMePh)₃ phosphine catalyst. Free energies were calculated at the (U)M06-2X/def2-TZVP/SMD(Toluene)//(U)M06-2X/def2-SVP level of theory and reported in kcal mol⁻¹. Reaction coordinates are not drawn to scale. **b**, NBO spin density difference plots of nucleophilic amination pathways. NBO charge and spin density values obtained from calculations performed at the same level of theory and basis set. Hydrogen atoms are omitted for clarity and bond lengths are reported in angstroms. **c**, *N*1-site selectivity of substrate **14** supported by kinetic barrier differences in the nucleophilic addition–intramolecular SET step. Calculations were performed at the (U)M06-2X/def2-TZVP/SMD(Toluene)//(U)

M06-2X/def2-SVP level of theory. Energy values are reported in kcal mol⁻¹, and reaction coordinates are not drawn to scale. **d**, General scheme and computational investigation of phosphine-dependent regioselectivity outcomes with 3-phenylpyrazole and methylene cyclopentane substrates. Reference ground states are set to distinct intermediates for comparison of the rate-determining step (RDS), and relative kinetic barriers and thermodynamic free energies for P(*p*-OMePh)₃ (black) and PCy₃ (grey) were calculated at the (U)M06-2X/def2-TZVP/SMD(Toluene)//(U)M06-2X/def2-SVP level of theory. Energy values are reported in kcal mol⁻¹. Hydrogen atoms are omitted for clarity and bond lengths are reported in angstroms. NCR, nitrogen-centred radical.

the ground state for the distonic radical cation (see Supplementary Table 53 for computational details), consistent with steric control on the transition state imparted by the differences in azole structure and subsequent approach to the distonic radical cation.

Last, we sought to understand the influence of phosphine identity on reaction regiochemical outcome (Fig. 4d). Our previous report for anti-Markovnikov selectivity indicated NCR addition to the alkene to be both rate- and selectivity-determining. Under the Markovnikov conditions, nucleophilic amination is proposed to be rate-determining. The selectivity switch between the anti-Markovnikov-selective $PCy_3$ and Markovnikov-selective $P(p\text{-OMePh})_3$ catalysts likely arises from differences in relative kinetic barriers of the two competing rate-determining steps. To probe this hypothesis, we calculated the potential energy surface of the Markovnikov nucleophilic amination pathway for $PCy_3$ as the phosphine catalyst and compared it with that arising with $P(p\text{-OMePh})_3$ when methylene cyclopentane was used as the alkene partner. When $P(p\text{-OMePh})_3$ is the catalyst, alkene addition is followed by a fast and irreversible nucleophilic amination through migratory insertion transition state **TS-1** ($\Delta G^{\ddagger} = +15.0$ kcal mol$^{-1}$, $\Delta G = -13.1$ kcal mol$^{-1}$) from a transient pentavalent phosphorus intermediate **INT-9**, leading to the Markovnikov product. In comparison, following azole addition and α-scission, NCR **INT-10** addition to the alkene through **TS-3** was calculated to be kinetically disfavoured by +2.2 kcal mol$^{-1}$ ($\Delta G^{\ddagger} = +17.2$ kcal mol$^{-1}$, $\Delta G = -15.7$ kcal mol$^{-1}$). However, with $PCy_3$, Markovnikov C–N bond formation was calculated to occur directly through nucleophilic addition –intramolecular SET of distonic radical cation **INT-4D'** through **TS-2'** ($\Delta G^{\ddagger} = +21.1$ kcal mol$^{-1}$, $\Delta G = -6.5$ kcal mol$^{-1}$), and was computed to be kinetically and thermodynamically less favourable than NCR addition by +3.9 kcal mol$^{-1}$, ultimately leading to anti-Markovnikov selectivity. Distortion–interaction analysis of **TS-1** and **TS-2'** suggests similar distortion and interaction effects for both transition states from the ground state distonic radical cation and azole (Supplementary Table 55). Therefore, it is likely that entropic penalties for **INT-4D'** to **TS-2'** contribute most significantly to the disparity in kinetic barriers. Furthermore, delocalization of the P–C phosphoranyl radical **INT-6A** into the aromatic π-system of $P(p\text{-OMePh})_3$ stabilizes the intermediate, and to a greater extent, **TS-1**, in contrast to what is possible with $PCy_3$ (ref. 55).

Based on these combined experimental and computational observations, we propose the following catalytic cycle for the Markovnikov hydroamination (Fig. 3g). Trivalent phosphine $PAr_3$ is oxidized through a photocatalytic reductive quenching cycle to **INT-3**. Alkene addition into **INT-3** generates distonic radical cation **INT-4**. **INT-4** undergoes the proposed nucleophilic addition–intramolecular SET (ii) or (ia) nucleophilic addition–(ib) migratory insertion C–N bond formation to **INT-6**. The P–C($sp^3$) bond of **INT-6** undergoes preferential homolysis to form **INT-7** and regenerate the trivalent phosphine. HAT with the thiol catalyst furnishes the *N*-alkylated product and simultaneously closes the photocatalytic cycle after electron transfer and proton transfer.

In conclusion, we leverage the reactivity of distonic phosphine radical cations with nucleophilic partners to develop a regioselective intermolecular hydroamination of a range of unactivated, terminal alkenes with numerous, distinct N–H azoles. The catalytic protocol complements existing radical-based anti-Markovnikov strategies and Markovnikov-selective, TM-catalysed methods that often face substrate limitations. Our mechanistic understanding of the phosphine-catalysed nucleophilic alkene functionalization step suggests avenues for extending the reactivity to the development of other valuable synthetic methods.

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

# Methods

Experimental and computational procedures for this study are available in the Supplementary Information.

## Data availability

Experimental procedures, characterization data and DFT calculations supporting the findings of this study are available in the Supplementary Information. X-ray crystallographic data are available free of charge from the Cambridge Crystallographic Data Centre, under reference no. 2476334.

**Acknowledgements** Financial support was provided by the National Institutes of Health (NIGMS R35 GM126986). A.J.M. acknowledges support from the NSF Graduate Research Fellowship Program (DGE-2034835). We acknowledge support from the NSF shared instrumentation grant CHE-1048804 and the NIH Office of Research Infrastructure Programs shared instrumentation grant S10OD028644. Computational and storage services associated with the Hoffman2 Shared Cluster were provided by the Research Technology Group of the UCLA Institute for Digital Research and Education. We thank T. J. Raab for assistance with X-ray structure determination. We also thank E. R. Wearing, A. Q. Cusumano and R. R. Knowles for helpful discussions and directions.

**Author contributions** F.F. designed, performed and analysed the experiments and computations. K.F.S. discovered the initial result. A.J.M. wrote the submission script template for DFT computations. A.G.D. designed and supervised the overall research project. F.F. and A.G.D. wrote the manuscript with inputs from all authors.

**Competing interests** The authors declare no competing interests.

**Additional information**
**Correspondence and requests for materials** should be addressed to Abigail G. Doyle.

## Extended Data Table 1 | Optimization studies

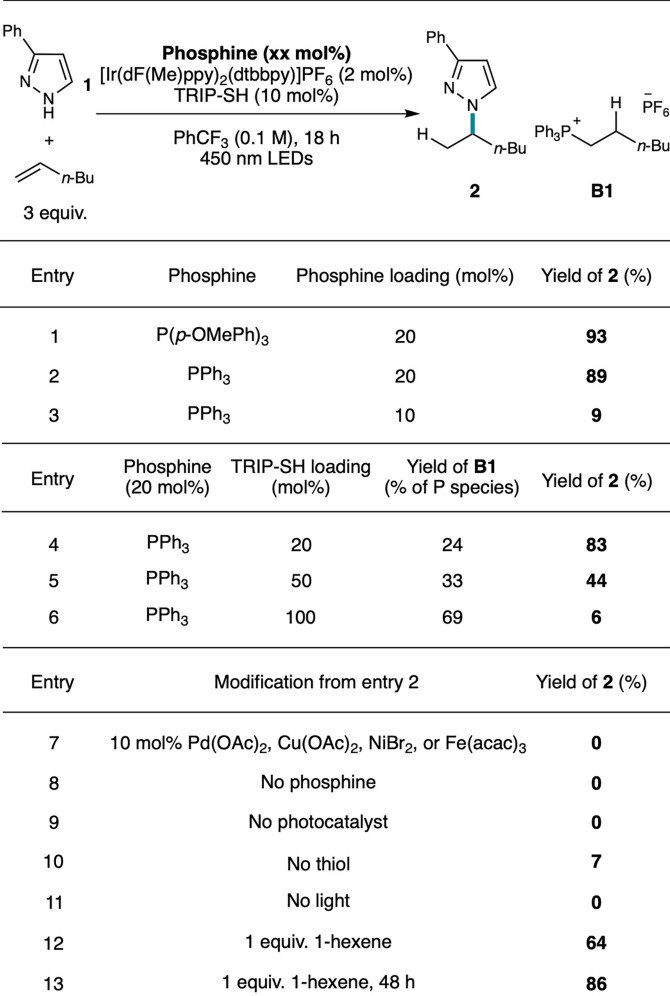

| Entry | Phosphine | Phosphine loading (mol%) | Yield of **2** (%) |
|---|---|---|---|
| 1 | P(*p*-OMePh)$_3$ | 20 | 93 |
| 2 | PPh$_3$ | 20 | 89 |
| 3 | PPh$_3$ | 10 | 9 |

| Entry | Phosphine (20 mol%) | TRIP-SH loading (mol%) | Yield of **B1** (% of P species) | Yield of **2** (%) |
|---|---|---|---|---|
| 4 | PPh$_3$ | 20 | 24 | 83 |
| 5 | PPh$_3$ | 50 | 33 | 44 |
| 6 | PPh$_3$ | 100 | 69 | 6 |

| Entry | Modification from entry 2 | Yield of **2** (%) |
|---|---|---|
| 7 | 10 mol% Pd(OAc)$_2$, Cu(OAc)$_2$, NiBr$_2$, or Fe(acac)$_3$ | 0 |
| 8 | No phosphine | 0 |
| 9 | No photocatalyst | 0 |
| 10 | No thiol | 7 |
| 11 | No light | 0 |
| 12 | 1 equiv. 1-hexene | 64 |
| 13 | 1 equiv. 1-hexene, 48 h | 86 |

Unless specified otherwise, reactions were performed on a 0.1 mmol scale reacting 1.0 equiv of 3-phenylpyrazole with 3.0 equiv of 1-hexene, PPh$_3$ (20 mol%), [Ir(dF(Me)ppy)$_2$dtbbpy]PF$_6$ (2 mol%), and TRIP-SH (10 mol%) irradiating with 450 nm LEDs for 18 h. Yields were determined by $^1$H NMR spectroscopic analysis against 1,3,5-trimethoxybenzene as an internal standard.