## [Peer Review File · Nature]

Markovnikov Hydroamination of Terminal Alkenes via Phosphine Redox Catalysis

Corresponding Author: Professor Abigail Doyle

Version 0:

Reviewer comments:

Referee #1

(Remarks to the Author)

The development of catalytic methods for C-N bond formation remains a central challenge in synthetic organic chemistry. Among these, the intermolecular hydroamination of unactivated alkenes represents an attractive yet elusive transformation due to issues of regioselectivity, functional group compatibility, and catalyst poisoning. In this manuscript, Doyle and coworkers describe a notable advancement in the field of hydroamination by introducing a phosphine/photoredox/hydrogen atom transfer (HAT) multicatalytic system that enables Markovnikov-selective intermolecular hydroamination of unactivated terminal alkenes with N-H azoles. The work is particularly significant as it addresses a long-standing challenge in synthetic organic chemistry using a main-group catalytic approach, complementing traditional transition-metal-based systems. The authors present a compelling mechanistic proposal involving a phosphine radical cation that mimics transition-metal-like reactivity, supported by a combination of experimental and computational evidence. While the study represents an important step forward, several aspects require further clarification and exploration to fully establish the generality, mechanism, and practical utility of the methodology. Addressing these challenges will not only enhance the practical utility of this method but also advance our fundamental understanding of phosphine radical cation chemistry.

1. Photocatalyst Scope and Rationale: The current study primarily employs iridium-based photocatalysts. Given the proposed mechanism, which involves oxidative quenching of the photocatalyst's excited state by the phosphine followed by reduction of the S radical, it would be instructive to explore whether other photosensitizers with comparable redox potentials, particularly organic dyes or earth-abundant metal complexes, could promote this transformation. A broader screening of photocatalysts, coupled with a discussion of the thermodynamic and kinetic requirements for each catalytic step, would strengthen the mechanistic framework of the method.

2. Expansion of Alkene Scope: The substrate scope is currently limited largely to terminal alkenes. It would be valuable to investigate whether other classes of alkenes, such as Michael acceptors, 1,1-disubstituted, 1,2-disubstituted, or 1,1,2-trisubstituted alkenes, are compatible with this system. Such studies would help delineate the steric and electronic constraints of the reaction and define its broader synthetic applicability.

3. Evaluation of Amination Reagents: The exclusive use of azoles as nitrogen nucleophiles raises the question of whether other commonly employed acyclic amines, such as diphenylamine, aniline, N-methylaniline, or benzamide, could participate in this transformation. Including examples of unsuccessful nucleophiles would provide a more realistic and informative scope, aiding other researchers in identifying suitable reaction systems.

4. Functional Group Tolerance: A systematic investigation of functional group compatibility (e.g., halides such as Br and I, electron-withdrawing groups like CF₃, CN, NO₂, and OCF₃) is warranted. Given that organocatalytic systems often exhibit complementary tolerance relative to metal-catalysed ones, such data would significantly bolster the claim of broad utility and functional group compatibility.

5. Selectivity in Indole and Pyrazole Functionalization: The moderate yields observed in the hydroamination of indoles (e.g., product 16, 42%) and certain azaindoles (17-20) warrant further discussion. In particular, analysis of potential byproducts, such as C3-alkylated indole derivatives, and a mechanistic explanation for the observed N1 versus C3 selectivity would be valuable. Additionally, the reactivity of unsubstituted pyrazole should be examined to evaluate the influence of substituents on reactivity and selectivity.

6. The addition of acetonitrile and water mixtures could help assess the potential formation of Ritter-type amination products, thereby providing insight into the intermediacy of carbocationic species and the role of the phosphine radical cation in

governing pathway selectivity.

7. To underscore the synthetic relevance of this methodology, its application in the formal synthesis of natural products or bioactive molecules would be highly compelling. Such examples would contextualize the utility of the method within complex molecular settings.

8. Including one or two application examples of this strategy in the formal synthesis of natural products or bioactive molecules would significantly enhance the practical value of this work.

9. Kinetic Isotope Effect (KIE) Studies: Performing KIE experiments using deuterated alkenes and/or deuterated azoles, in conjunction with additional kinetic analyses, could provide critical insight into the rate-determining step and further validate the proposed mechanism.

10. To rule out a radical chain mechanism, light-on/light-off experiments and measurement quantum yield are recommended, although such a pathway seems unlikely based on the current data.

11. To elucidate the origin of Markovnikov selectivity observed in terminal alkene reactions, the authors computationally investigated the addition of INT-3 to 1-hexene. The calculations revealed that the anti-Markovnikov product is kinetically favored ($\Delta\Delta G^\ddagger = +1.7$ kcal/mol), with the elementary step likely being irreversible under the reaction conditions. However, the manuscript lacks direct comparison between the forward and reverse reaction barriers. Specifically, we could not locate any figures illustrating these energy barriers apart from the textual description. Please indicate which figure in the manuscript corresponds to this analysis.

12. Regarding the C–N bond formation process, the authors proposed and discussed two distinct reaction mechanisms. Computational results demonstrate that the energy barrier for the TS1 pathway is lower than that of TS2. Although both nucleophilic amination pathways could potentially be operative, the reaction is predicted to preferentially proceed via the TS1 transition state.

Referee #2

(Remarks to the Author)

The present work by Doyle and co-workers describes a photoredox-catalyzed manifold for the Markovnikov-selective hydroamination of terminal alkenes with azoles. The protocol is based on a key organophosphorus co-catalyst that, through a P(IV) radical species, unlocks named transformation, which is typically restricted to transition-metal based catalysis.

I find the work solid and robust under the preparative point of view; however, the most important part of the ms is the in-depth mechanistic analysis reported, which lays the foundation for the entire work and justifies the arguments offered by the Authors.

Overall, I am completely satisfied with the work: both its novelty and importance render it suitable for publication in Nature. The quality of the current version of the ms is really high.

There are, however, a couple of elements that have been indicated, but not fully detailed/explained in my opinion. First of all, it should be clearly indicated in the title that the reaction only applies to terminal alkenes. At the same time, it would be interesting to know what happens with internal alkenes (e.g., di-, tri- and tetra-substituted derivatives). Are these competent substrates? Is the reactivity diverted with such substrates or are they unreactive? If relevant, how does the mechanism change with such substrates? What about the adoption of nucleophiles different from azoles?

Referee #3

(Remarks to the Author)

The manuscript by Doyle and co-workers describes the addition of nitrogen nucleophiles to alkenes through the combined effect of phosphine and thiol catalysts in conjunction with an Ir photocatalyst. The remarkable finding in this paper is that by altering the phosphine catalyst, the regiochemical outcome of the reaction is inverted – whereas PCy₃ catalyst gives products of N-addition to the less substituted carbon (anti-Markovnikov), with triarylphosphines, the nitrogen adds to the more substituted carbon. Thus, a fundamentally new reaction mechanism operates that involves an unanticipated and, until now, not yet observed elementary step. This is an exciting development, especially since the intimate association of the phosphine catalyst with the substrate immediately suggests many opportunities for catalyst control of regio and stereoselectivity that is not easily obtainable with other reactivity modes (i.e. by direct N-centered radical addition to the alkene). In addition to the fact that this paper can be expected to enable new modes of main group catalysis, the transformation that is accomplished is an important one. I enthusiastically support publication of this work in Nature subject to addressing the following minor points:

(1) A brief comment should be included about the reason glovebox/Schlenk techniques are needed in this type of catalysts. The Ir, PR₃, and thiol catalysts don't seem highly sensitive to oxygen, but I assume the reaction can't simply be spared with N₂ prior to irradiation or the authors would have done this.

(2) Some mechanistic rationale should be provided for the superior reactivity of PPh₃ for some substrates and P(p-MeOPh)₃ for others.

(3) Line 235-236: I suspect B1 and B2 have been switched.

(4) Line 327: I was puzzled why there is selectivity arising from the location of the unpaired spin density relative to the pyridine nitrogen, but then later in the paragraph it seems selectivity arises simply from a steric effect. Perhaps rephrase so the reader doesn't unnecessarily puzzle over an electronic effect.

(5) Page 73 of the SI, the TS drawing for TS-2-N3 and TS-2-N1 are identical.

(6) The calculations are well done and thorough, but it may be helpful to add a calculation for the PRC to react with the alkene and the azole. If reaction with the PRC is exergonic enough, it is possible that this step influences which pathway is

followed.

Version 1:

Reviewer comments:

Referee #1

(Remarks to the Author)

The author has diligently addressed the points raised by the reviewer, leading to significant improvements in the quality and completeness of the manuscript. It now meets the high standards expected by Nature. This reviewer recommends that the excellent paper be accepted in its current form.

Referee #2

(Remarks to the Author)

The authors have performed a thorough and detailed revision, essentially addressing all the issues raised by the reviewers. Acceptance in the present form is warmly recommended.

Referee #3

(Remarks to the Author)

The authors have addressed the concerns I had with the earlier version of this manuscript and, in my opinion, it can be accepted for publication.

Editor:

We do not need you to make bioactive molecules or natural products via this method (as per their comments 7/8). However, the questions raised about FG tolerance/scope, explicitly noting the limitation to terminal alkenes, queries about using other nucleophiles are all reasonable (and in many cases echoed by the other reviewers).

Thank you for your guidance in the review process. Please see below for our additions in response to the Reviewer's suggestions and questions.

Reviewer 1:

1. Photocatalyst Scope and Rationale: The current study primarily employs iridium-based photocatalysts. Given the proposed mechanism, which involves oxidative quenching of the photocatalyst's excited state by the phosphine followed by reduction of the S radical, it would be instructive to explore whether other photosensitizers with comparable redox potentials, particularly organic dyes or earth-abundant metal complexes, could promote this transformation. A broader screening of photocatalysts, coupled with a discussion of the thermodynamic and kinetic requirements for each catalytic step, would strengthen the mechanistic framework of the method.

We agree with the reviewer that organic photosensitizers or earth-abundant metal complexes would be valuable to promote this transformation. In Table S5, we have included screening data with additional organic photocatalysts that possess reduction potentials to match that of the iridium-based photocatalyst currently employed, but these were found to be ineffective. We attribute the poor performance of these organic photocatalysts to a poorly matched ability to reduce the thiyl radical (thermodynamic requirement); these photocatalysts are also known to be more susceptible to back-electron transfer (kinetic requirement), which we found to be a major limitation in our prior work on anti-Markovnikov hydroamination with phosphoranyl radicals. We have included a discussion and analysis of these results in the Supporting Information, as requested by the reviewer.

2. Expansion of Alkene Scope: The substrate scope is currently limited largely to terminal alkenes. It would be valuable to investigate whether other classes of alkenes, such as Michael acceptors, 1,1-disubstituted, 1,2-disubstituted, or 1,1,2-trisubstituted alkenes, are compatible with this system. Such studies would help delineate the steric and electronic constraints of the reaction and define its broader synthetic applicability.

We appreciate the reviewer's question about the compatibility of other classes of alkenes under these conditions. We have explored all of the classes mentioned above and included new results in the manuscript and Supporting Information (Section 9) to help readers understand the broader synthetic applicability. To summarize, we found that Michael acceptors such as *tert*-butyl acrylate and acrylonitrile were unreactive when employed as the alkene partner. We suspect that this may be due to ineffective addition of the alkene to the phosphine radical cation due to the alkene's reduced nucleophilicity or competitive nucleophilic addition of the phosphine to the electron-deficient alkene, an elementary step common to Baylis–Hillman reactions.

In general, internal acyclic alkenes, either 1,1-disubstituted, 1,2-disubstituted, 1,1,2-trisubstituted, or tetra-substituted, are less reactive substrates than terminal alkenes. In the scope, the internal 1,1,2-substituted alkene and enone of products **36** and **37** are untouched, highlighting the chemoselectivity for terminal alkenes.

We have added 1,1-diphenylethylene and tetramethylethylene in the SI as additional alkenes that gave poor reactivity (<10% reactivity), which we hypothesize is due to slow nucleophilic addition of the azole to the hindered distonic phosphine radical cation. With 1,1-diphenylethylene, the reaction is Markovnikov selective, albeit low yielding, consistent with the reaction outcome using terminal alkenes.

In the manuscript, we have added a reaction with a tri-substituted alkene (2-methylbut-2-ene). With this alkene, we observe an anti-Markovnikov to Markovnikov ratio of 1.6:1 and quite low overall yield (14%) with $P(p\text{-OMePh})_3$ as catalyst. By comparison, when PCy_3 is employed as phosphine catalyst, the reaction gives a higher product yield and a regioisomeric ratio that more strongly favors the anti-Markovnikov product. There are two possible interpretations of this result that we have considered. In one, nucleophilic amination is poorly selective since the two regioisomeric distonic radical cations are likely close in energy, but the reactivity is poor due to steric hindrance of forming the distonic radical cation and fully substituted C–N bond. Alternatively, the outcome with $P(p\text{-OMePh})_3$ as catalyst may be a competition between the two pathways (nucleophilic amination of the distonic radical cation versus NCR addition to the alkene), leading to low selectivity.

A 1,2-disubstituted substrate such as β -methylstyrene is largely unreactive under both anti-Markovnikov conditions (PCy_3 as phosphine catalyst) and Markovnikov conditions ($P(p\text{-OMePh})_3$ as phosphine catalyst), likely because styrenes and terminal alkenes are unreactive under the anti-Markovnikov conditions and internal alkenes are sterically congested for this Markovnikov chemistry. However, we observe *N*-alkylation at the α -position when $P(p\text{-OMePh})_3$ is used as catalyst, and *N*-alkylation at the β -position when PCy_3 is used as catalyst.

We have included these additional alkenes to illustrate the breadth and limitations of the method, and included the following discussion in the main text, from line 186:

Whereas internal alkenes (41) and (42) displayed low reactivity, the hydroamination regioselectivity was strongly phosphine-dependent, with P(*p*-OMePh)₃ delivering greater Markovnikov or exclusively Markovnikov selectivity, respectively, compared to that obtained with PCy₃ in these challenging case studies. Overall, this hydroamination method effectively couples together a range of N–H azoles and functionalized terminal alkenes with exclusive *N*-site selectivity and Markovnikov regioselectivity, encompassing substrates both compatible and incompatible with transition-metal catalysis under a general set of conditions.

Although some of these alkene classes are low yielding, the selectivity observed suggests that further phosphine optimization could enable the development of highly selective reactions for what are otherwise very challenging examples for selective olefin functionalization.

*3. Evaluation of Amination Reagents: The exclusive use of azoles as nitrogen nucleophiles raises the question of whether other commonly employed acyclic amines, such as diphenylamine, aniline, *N*-methylaniline, or benzamide, could participate in this transformation. Including examples of unsuccessful nucleophiles would provide a more realistic and informative scope, aiding other researchers in identifying suitable reaction systems.*

We agree with the reviewer that acyclic amines warrant further exploration under these catalytic conditions. In Table S13, we have included screening with acyclic nucleophiles such as *p*-*tert*-Bu-aniline and benzamide, which were unreactive in this protocol. Per the reviewer's request, we have additionally screened diphenylamine and *N*-methylaniline; both were found to be unreactive. We also tested these nucleophiles as additives, which addresses the next comment by the reviewer (see point 4); since these amines are preserved at the end of the reaction, we hypothesize that they act as competitive quenchers of the excited photocatalyst and thus result in reduced reactivity of the desired reaction, which is initiated by single electron oxidation of the phosphine by the excited photocatalyst. We have included examples of these unsuccessful nucleophiles in the new functional group robustness screen section of Figure 2 and have included the rest of this data in the Supporting Information. We hypothesize that cyclic amines are currently required for reactivity due to a combination of conformational effects with possible non-covalent interactions between these structures and the distonic radical cation, and increased charge delocalization / better electron transfer for nucleophilic amination. Although acyclic amines are not reactive, we discovered that reactivity does extend to other cyclic nitrogen nucleophiles and have included these examples in the scope:

Interestingly, carbazole (21), oxindole (22), a partially saturated N–H azole, thio- (23) and benzo- (24) ureas, and fully saturated carbamate (25), were also reactive in this hydroamination method, demonstrating that nitrogen nucleophiles other than unsaturated azoles are compatible with the catalytic protocol.

4. *Functional Group Tolerance: A systematic investigation of functional group compatibility (e.g., halides such as Br and I, electron-withdrawing groups like CF₃, CN, NO₂, and OCF₃) is warranted. Given that organocatalytic systems often exhibit complementary tolerance relative to metal-catalysed ones, such data would significantly bolster the claim of broad utility and functional group compatibility.*

We agree with the reviewer that a systematic investigation of functional group compatibility can offer a more comprehensive understanding of the reaction scope and limitations. We have performed an additive screen and included this data (see below) in the manuscript and SI (Section 3). As expected, functional groups such as aniline, phenol, and nitroarene that can undergo competitive oxidation or reduction by the photocatalyst were deleterious to the standard reaction. However, aryl bromides and chlorides, which are coupling partners in transition-metal catalyzed methods, were tolerated under the reaction conditions and fully recovered at the end of the reaction, highlighting a benefit of using a main group element catalyst. Functional groups like trifluoromethoxy and triflate had a moderate influence on the standard reaction yield but were compatible under these conditions. Benzoates, pyridines, alcohols, and carbamates were largely unreactive under these conditions, resulting in a minor influence on the yield of the standard reaction.

Yields and additive recovery determined by ¹H NMR in CDCl₃ with 1,3,5-TMB as internal standard.

5. *Selectivity in Indole and Pyrazole Functionalization: The moderate yields observed in the hydroamination of indoles (e.g., product 16, 42%) and certain azaindoles (17-20) warrant further discussion. In particular, analysis of potential byproducts, such as C3-alkylated indole derivatives, and a mechanistic explanation for the observed N1 versus C3 selectivity would be valuable. Additionally, the reactivity of unsubstituted pyrazole should be examined to evaluate the influence of substituents on reactivity and selectivity.*

We thank the reviewer for bringing this consideration to our attention. For indole *N*-alkylated product **16**, the remaining mass balance is isolated unreacted indole starting material and byproduct resulting from INT-

9 addition into excess hexene. Letting the reaction run for longer (36 h instead of 18 h) did not lead to improved yield.

We did not observe any C3-alkylated indole derivatives, which we hypothesize is due to the conformational unfavourability of C3 addition into the distonic radical cation, as computational attempts to locate a transition state complex were unsuccessful. Alternatively, we considered indole addition to proceed through a P(V) intermediate of the INT-11 type. We found a stable ground-state complex and “migratory insertion” transition state with the indole nitrogen (see SI, Section 10), in line with the proposed nucleophilic amination pathway.

The standard reaction with pyrazole proceeded to give 57% yield of Markovnikov product.

The presence of the phenyl substituent may provide additional non-covalent stabilization that aids pyrazole reactivity but is certainly not required for reactivity.

6. The addition of acetonitrile and water mixtures could help assess the potential formation of Ritter-type amination products, thereby providing insight into the intermediacy of carbocationic species and the role of the phosphine radical cation in governing pathway selectivity.

We thank the reviewer for bringing this consideration to our attention. Since our proposal is that amination does not proceed through a carbocation intermediate, we would anticipate that no Ritter products would be formed; unless acetonitrile, like the azole nucleophiles, can add into the carbon-centered radical and promote radical migration. In Table S10 of the Supporting Information, we screened acetonitrile as a solvent and observed significantly reduced reactivity, resulting in a 9% yield of azole Markovnikov product. The reduced reactivity with acetonitrile may arise because the solvent can serve as a hydrogen atom source to quench the distonic radical cation. The same reaction containing an additional 2 equivalents of water also gave 9% yield of azole product; we did not observe any Ritter amination products. Performing the standard reaction in acetonitrile followed by aqueous workup also did not result in any Ritter-type products. The reduced reactivity with water is likely because water can act as a nucleophile toward phosphine radical cations¹ and undergo β -scission. Consistent with this reactivity, we observe complete consumption of triphenylphosphine in the latter reaction with only phosphonium **B1** and phosphine oxide peaks in ³¹P NMR. We also repeated our standard reaction, adding in 1 equivalent each of acetonitrile and water; we observe similarly low (4%) yield of azole Markovnikov product and no Ritter amination products.

7. To underscore the synthetic relevance of this methodology, its application in the formal synthesis of natural products or bioactive molecules would be highly compelling. Such examples would contextualize the utility of the method within complex molecular settings.
8. Including one or two application examples of this strategy in the formal synthesis of natural products or bioactive molecules would significantly enhance the practical value of this work.

We thank the reviewer for the suggestion. Given the emphasis of this manuscript on the discovery and interrogation of a mechanistically novel activation mode for main group element catalysis, we believe that an application to the preparation of a complex target is outside the scope of the current study.

9. Kinetic Isotope Effect (KIE) Studies: Performing KIE experiments using deuterated alkenes and/or deuterated azoles, in conjunction with additional kinetic analyses, could provide critical insight into the rate-determining step and further validate the proposed mechanism.

We thank the reviewer for this suggestion. We have proposed C–N bond formation to be rate-determining, and have postulated two possibilities for this step; either (i) migratory insertion from P(V) intermediate INT-11 or (ii) nucleophilic addition into phosphine distonic radical cation INT-4.

As suggested by the reviewer, we obtained a KIE value of 1.032 ± 0.647 using N–D deuterated pyrazole (see SI, Section 9). Given the error in measurements, likely from the impracticality of making stock solutions with insoluble reagents and the variability in our light setup, we interpret this value as the absence of a KIE, which is in line with our hypothesis that N–H bond-breaking is not involved in the RDS. Independent rate experiments with deuterated alkene, styrene- α -d₁, also revealed higher yields of deuterated product compared with yields of standard product **40**; however, similar challenges in reaction setup limit the derivation of an accurate KIE value. We considered using Singleton's method to obtain KIEs; however, the proposal of multiple mechanisms involved in the catalytic cycle together with experimental evidence of byproduct formation is likely to hinder the utility of this method.

Since the alkene partner is in excess under standard conditions, we pursued a competition experiment with styrene- α -d₁ and observed an inverse secondary kinetic isotope effect ($k_{\text{H}}/k_{\text{D}} = 0.95$, see SI, Section 9), consistent with a sp² to sp³ change in hybridization that would occur during the proposed rate-determining C–N bond formation.

We calculated the predicted KIE for C–N bond formation from a P(V) intermediate for 3-phenylpyrazole with styrene and $P(p\text{-OMePh})_3$ as phosphine, also finding an inverse secondary KIE ($k_{\text{H}}/k_{\text{D}} = 0.934$) that matches with our experimental value. Procession of C–N bond formation via either pathway appears to be substrate-dependent. Therefore, we also investigated several other substrate pairings computationally. The predicted KIE for C–N bond formation in **TS-1** (for 3-phenylpyrazole with 1-hexene and $P(p\text{-OMePh})_3$ as phosphine) and in **TS-2** (for 4-azabenzimidazole (*N*1-alkylation) with 1-hexene and $P(p\text{-OMePh})_3$ as phosphine) are $k_{\text{H}}/k_{\text{D}} = 0.777$ for **TS-1** and $k_{\text{H}}/k_{\text{D}} = 0.760$ for **TS-2**, respectively. Taken together, this range of KIE values further support C–N bond formation as the rate-determining step. However, given the similarity of the predicted KIEs between the two transition states and errors in measurement, we do not think we can use the experimental value to distinguish between **TS-1** and **TS-2**.

Separately, with N–D pyrazole as the azole substrate, the associated spectra collected indicate the presence of deuterium atom transfer at the terminal carbon in the *N*-alkylated product (see SI, Section 9), further validating the proposed terminal alkyl radical as a productive intermediate.

10. To rule out a radical chain mechanism, light-on/light-off experiments and measurement quantum yield are recommended, although such a pathway seems unlikely based on the current data.

We agree with the reviewer that these experiments can help to rule out a radical chain mechanism. We have performed the suggested light-on/light-off experiments and measured quantum yield ($\text{QY} = 0.00017$) for the standard reaction, the results of which indicate the reaction is not proceeding via a radical chain mechanism. This information can be found in the Supporting Information (Section 9).

Fig. S5. Light-on light-off experiments.

11. To elucidate the origin of Markovnikov selectivity observed in terminal alkene reactions, the authors computationally investigated the addition of INT-3 to 1-hexene. The calculations revealed that the anti-Markovnikov product is kinetically favored ($\Delta\Delta G_{\ddagger}^{\ddagger} = +1.7$ kcal/mol), with the elementary step likely being irreversible under the reaction conditions. However, the manuscript lacks direct comparison between the forward and reverse reaction barriers. Specifically, we could not locate any figures illustrating these energy barriers apart from the textual description. Please indicate which figure in the manuscript corresponds to this analysis.

We thank the reviewer for bringing this consideration to our attention. We have now noted in the main text the location of the figure illustrating the direct comparison between the forward and reverse reaction barriers in the Supporting Information (Fig. S40).

Fig. S40. Regioselective outcomes for addition of 3-phenylpyrazole NCR to 1-hexene.

12. Regarding the C–N bond formation process, the authors proposed and discussed two distinct reaction mechanisms. Computational results demonstrate that the energy barrier for the TS1 pathway is lower than that of TS2. Although both nucleophilic amination pathways could potentially be operative, the reaction is predicted to preferentially proceed via the TS1 transition state.

We agree with the reviewer that two distinct reaction mechanisms for nucleophilic amination are proposed and discussed. Our conclusion is that both are potentially feasible and the favored pathway likely varies as a function of the nucleophile. As the reviewer pointed out, for the pairing of 3-phenylpyrazole with methylene cyclopentane (Fig. 6A), we find the energy barrier for the TS1 pathway to be lower than that of TS2, and the reaction is predicted to preferentially proceed via the migratory insertion transition state TS1. However, for 4-azabenzimidazole, we can only find a transition state corresponding to TS2, and this pathway explains the observed N-site selectivity observed experimentally.

Reviewer 2:

1. First of all, it should be clearly indicated in the title that the reaction only applies to terminal alkenes.

We thank the reviewer for bringing this consideration to our attention and have changed the title accordingly to: Phosphine/Photoredox-Catalyzed Markovnikov Amination of Terminal Alkenes with Azoles.

2. It would be interesting to know what happens with internal alkenes (e.g., di-, tri- and tetra-substituted derivatives). Are these competent substrates? Is the reactivity diverted with such substrates or are they unreactive? If relevant, how does the mechanism changes with such substrates?

We agree with the reviewer for bringing up this question. We have addressed this comment in response to comment #2 from reviewer 1 (see SI, Section 9), and have added the following alkenes, paired with 3-phenylpyrazole as the N–H azole, into the manuscript:

3. What about the adoption of nucleophiles different from azoles?

The question from the reviewer is appreciated. A list of nucleophiles we have attempted can be found in Table S13 and in Section 9 of the Supporting Information. We have commented on alternative nucleophiles in response to comment #3 from reviewer 1 and discovered a handful of new nucleophile classes that are effective under the current catalytic conditions (shown below). That said, it does seem that saturated and unsaturated azoles are privileged nucleophiles for this method. Fortunately, this class comprises many interesting and valuable structural motifs. In our future studies, we will attempt to understand the limitation mechanistically and use this information to inform new synthetic advances.

Reviewer 3:

1. A brief comment should be included about the reason glovebox/Schlenk techniques are needed in this type of catalysts. The Ir, PR₃, and thiol catalysts don't seem highly sensitive to oxygen, but I assume the reaction can't simply be spared with N₂ prior to irradiation or the authors would have done this.

The comment from the reviewer is appreciated. Setting up the standard reaction with PPh₃ under ambient conditions without sparging led to no desired product; sparging the reaction with N₂ for 5 minutes prior to irradiation led to 55% yield (76% yield with P(*p*-OMePh)₃), demonstrating that moderate to good reactivity can be achieved without rigorous air-exclusion procedures, but the use of a glovebox and/or Schlenk techniques facilitate higher yield. We have included these yields in Figure 2 and have modified the text from line 151 to read:

While the standard reaction conditions call for the use of a glovebox, setting up the reaction under ambient conditions followed by N₂ sparging or using standard Schlenk technique demonstrated similar reactivity; products **2** and **6** were obtained in 76 and 68% yield with N₂ sparging and Schlenk technique versus 85 and 75% yield using the glovebox, respectively.

*2. Some mechanistic rationale should be provided for the superior reactivity of PPh₃ for some substrates and P(*p*-MeOPh)₃ for others.*

We thank the reviewer for bringing this consideration to our attention. As indicated in our Stern-Volmer studies (see Fig. S4), some azoles, such as indole, can outcompete PPh₃ in quenching of the photocatalyst. Therefore, for these substrates, a more effective phosphine quencher, such as the more electron-rich P(*p*-OMePh)₃ (see Fig. S2), is required to achieve reactivity and/or higher reactivity (Table S13). An exception is that 1,2,4-triazole is only reactive with 1-hexene when PPh₃ is used as the phosphine catalyst; one possibility is that a less electron-rich phosphine is expected to better activate the distonic radical cation to nucleophilic addition with this azole.

Fig. S4. Comparison of I₀/I vs. [quencher] for PPh₃ and N-H azole substrates.

Fig. S2. Plot of I₀/I with (*p*-OMePh)₃P and PPh₃ quenchers.

We have added this rationale to the main text, following line 239:

P(*p*-OMePh)₃, a more electron-rich phosphine, is required when employing azole substrates that can undergo competitive oxidation by the excited-state photocatalyst (see supplementary materials, Fig. S4).

3. Line 235-236: I suspect B1 and B2 have been switched.

We thank the reviewer for bringing this consideration to our attention and have corrected this numbering.

4. Line 327: I was puzzled why there is selectivity arising from the location of the unpaired spin density relative to the pyridine nitrogen, but then later in the paragraph it seems selectivity arises simply from a steric effect. Perhaps rephrase so the reader doesn't unnecessarily puzzle over an electronic effect.

We agree with the reviewer that the current text could be made clearer. To avoid confusion, we have removed the reference to the rationale for selectivity observed under the anti-Markovnikov conditions, since it is irrelevant to our proposed mechanism for Markovnikov regioselectivity. The paragraph, starting from line 342, now reads as:

To validate the feasibility of this pathway, we questioned whether our calculations could rationalize the complete *NI*-site selectivity observed experimentally. Indeed, under the proposed

C–N bond formation manifold, the computed activation barriers for *NI* vs *N3* alkylation reveal a $\Delta\Delta G^\ddagger$ of 4.9 kcal/mol favoring *NI* alkylation (Fig. 6B). Distortion-Interaction analysis of the two transition states suggests there to be greater interaction between the azole and distonic radical cation for *NI* nucleophilic attack, as well as reduced distortion from the ground state for the distonic radical cation (see supplementary materials, Table S53 for computational details), consistent with steric control on the transition state imparted by the differences in azole structure and subsequent approach to the distonic radical cation.

5. Page 73 of the SI, the TS drawing for TS-2-N3 and TS-2-N1 are identical.

We thank the reviewer for bringing this consideration to our attention and have corrected this figure.

6. The calculations are well done and thorough, but it may be helpful to add a calculation for the PRC to react with the alkene and the azole. If reaction with the PRC is exergonic enough, it is possible that this step influences which pathway is followed.

We agree with the reviewer that computations for the azole and alkene addition into the phosphine radical cation would be useful to provide additional support for the proposed mechanism. We have included further discussion of this comparison in the Supporting Information (Section 10). In short, computational and experimental evidence suggest that if the azole substrate is reactive, both the barriers for alkene and azole addition into the phosphine radical cation are lower than either NCR addition into the alkene (anti-Markovnikov selectivity) and nucleophilic amination into the distonic radical cation (Markovnikov selectivity) (Figure 6C), with downstream products resulting from alkene and azole addition into the phosphine radical cation being observed experimentally.

Sincerely,

Abby

References

1. Zhang, J., Mück-Lichtenfeld, C. & Studer, A. Photocatalytic phosphine-mediated water activation for radical hydrogenation. *Nature* **619**, 506–513 (2023).
2. Chinn, A. J., Sedillo, K. & Doyle, A. G. Phosphine/Photoredox Catalyzed Anti-Markovnikov Hydroamination of Olefins with Primary Sulfonamides via α -Scission from Phosphoranyl Radicals. *J. Am. Chem. Soc.* **143**, 18331–18338 (2021).
3. Liu, K. *et al.* Photon-driven radical hydro-phosphoniumylation of unactivated olefins. *Chem Catal.* **5**, (2024).
4. Ando, T., Yokogawa, D., Ohmatsu, K. & Ooi, T. Deoxygenative [3 + 2] Annulation of α,β -Unsaturated Carbonyl Compounds and Electron-Rich Olefins via Photocatalytic Umpolung of Triarylphosphine. *J. Am. Chem. Soc.* (2025) doi:10.1021/jacs.5c06114.
5. Sedillo, K., Fan, F., Knowles, R. R. & Doyle, A. G. Cooperative Phosphine-Photoredox Catalysis Enables N–H Activation of Azoles for Intermolecular Olefin Hydroamination. *J. Am. Chem. Soc.* **146**, 20349–20356 (2024).